# Synthesis, Characterization, and Application of Magnetic Zeolite Nanocomposites: A Review of Current Research and Future Applications

**DOI:** 10.3390/nano15120921

**Published:** 2025-06-13

**Authors:** Sabina Vohl, Irena Ban, Janja Stergar, Mojca Slemnik

**Affiliations:** University of Maribor, Faculty of Chemistry and Chemical Engineering, Smetanova 17, 2000 Maribor, Slovenia; sabina.vohl@um.si (S.V.); irena.ban@um.si (I.B.); mojca.slemnik@um.si (M.S.)

**Keywords:** magnetic nanoparticles, zeolites, magnetic zeolite nanocomposites, adsorptive removal of pollutants

## Abstract

Magnetic zeolite nanocomposites (NCs) have emerged as a promising class of hybrid materials that combine the high surface area, porosity, and ion exchange capacity of zeolites with the magnetic properties of nanoparticles (NPs), particularly iron oxide-based nanomaterials. This review provides a comprehensive overview of the synthesis, characterization, and diverse applications of magnetic zeolite NCs. We begin by introducing the fundamental properties of zeolites and magnetic nanoparticles (MNPs), highlighting their synergistic integration into multifunctional composites. The structural features of various zeolite frameworks and their influence on composite performance are discussed, along with different interaction modes between MNPs and zeolite matrices. The evolution of research on magnetic zeolite NCs is traced chronologically from its early stages in the 1990s to current advancements. Synthesis methods such as co-precipitation, sol–gel, hydrothermal, microwave-assisted, and sonochemical approaches are systematically compared, emphasizing their advantages and limitations. Key characterization techniques—including X-Ray Powder Diffraction (XRD), Fourier Transform Infrared Spectroscopy (FTIR), Scanning and Transmission Electron Microscopy (SEM, TEM), Thermogravimetric Analysis (TGA), Nitrogen Adsorption/Desorption (BET analysis), Vibrating Sample Magnetometry (VSM), Zeta potential analysis, Inductively Coupled Plasma Optical Emission Spectroscopy (ICP-OES), and X-Ray Photoelectron Spectroscopy (XPS)—are described, with attention to the specific insights they provide into the physicochemical, magnetic, and structural properties of the NCs. Finally, the review explores current and potential applications of these materials in environmental and biomedical fields, focusing on adsorption, catalysis, magnetic resonance imaging (MRI), drug delivery, ion exchange, and polymer modification. This article aims to provide a foundation for future research directions and inspire innovative applications of magnetic zeolite NCs.

## 1. Introduction

Zeolites are crystalline aluminosilicate minerals with intricate three-dimensional frameworks formed by the low-temperature transformation of volcanic rocks. They are found in diverse geological settings, primarily as altered authigenic minerals, low-temperature and low-pressure minerals in metamorphic systems, and secondary minerals in weathered zones. Nearly 300 different types of natural zeolites are known, with clinoptilolite, chabazite, erionite, mordenite, and phillipsite being the most commercially valuable [1]. The Swedish chemist Axel Cronstedt was the first to identify zeolites in 1756. He observed that the mineral stilbite seemed to boil when heated, which led him to name these substances “zeolites” or “boiling stones” [2]. Chemically, zeolites are microporous aluminosilicate minerals that have a crystal structure with a system of interconnected tunnels and cages. Zeolites are aluminosilicates characterized by exceptional physicochemical properties; in particular, their cation exchange capacity, extensive surface area, molecular sieving capabilities, catalytic activity, and sorption potential tributes underpin their widespread use as adsorbents and catalysts in various industrial applications [3]. Well-known hydrated aluminosilicate minerals can be sourced from natural deposits or produced synthetically using various methods and raw materials, with their crystalline structure influenced by factors like the chemical composition of the reaction mixture, synthesis time, temperature, and structure-directing agents. Typically produced as powders, their practical application is hindered by the need for pelletization, which reduces internal surface area and performance. To address this, strategies such as creating hierarchical structures or forming nanocomposites (NCs) with other materials have been explored [4]. NC materials have been extensively studied for over three decades due to their current and potential functional applications. These materials stand out for their high surface-to-volume ratio and significant interfacial area of embedded nanoparticles (NPs). Additionally, the size-dependent properties of NPs allow for the enhancement of host materials with diverse functionalities. Magnetic nanoparticles (MNPs) are among the most widely used functional fillers. These NPs serve dual purposes: they can function independently or as part of composite materials. However, the magnetic properties of NCs are often influenced by factors such as particle dispersion, interparticle interactions and surface effects on the NPs’ magnetism [5]. NCs offer the advantage of multifunctionality, allowing zeolite-based materials to incorporate additional properties. In magnetic zeolite NCs, the integration aims to preserve the zeolite’s structural integrity while imparting magnetic properties. Such composites are particularly valuable for simplifying recovery processes, such as separating zeolites from liquid phases, which would otherwise require costly techniques [4]. Materials used for magnetic purposes in composites include (Fe, Co, Ni, …), ferrites (MFe_2_O_4_; M = Mn^2+^, Co^2+^, Ni^2+^, etc.), alloys (CoPt_5_, FePt, …), and iron oxides (Fe_3_O_4_, γ-Fe_2_O_3_).

Nanotechnology has garnered global attention, with significant efforts directed towards building foundational knowledge to manipulate and restructure materials at the nanoscale. This field enables the creation of innovative functional materials with sizes ranging from 1 to 100 nm, exhibiting unique properties distinct from their bulk counterparts [6]. In recent years, considerable efforts have focused on synthesizing NCs to develop high-performance materials, owing to their exceptional properties and versatile design potential. NCs are solids composed of two distinct phases, one of which features dimensions at the nanometer scale [7].

MNPs are recognized for their non-toxic nature, high recyclability, reusability, and ease of separation under an external magnetic field [8]. Incorporating NPs into solid substrates further improves their chemical stability and biocompatibility, reduces aggregation, and allows for better control over their size and shape [9].

Among cost-effective materials, zeolites stand out due to their remarkable properties, including a high surface area, unique micropore structures, diverse channel systems, and strong resistance to chemical and thermal treatments [7]. Zeolites are highly beneficial materials in different fields, especially as adsorbents, ion exchangers, and catalysts [10]. A significant challenge is the separation of fine zeolite particles after the adsorption process, as conventional methods often struggle to efficiently remove these particles from treated water. This issue can be mitigated by synthesizing zeolites with magnetic properties, enabling their removal from aqueous systems through the application of a magnetic field [11]. Additionally, the ability to regenerate magnetic zeolites enhances their value for a wide range of applications [12].

Technological advancements are increasingly being leveraged to tackle environmental challenges. Magnetic particle technology has gained significant attention in recent years due to its dual functionality in adsorption and separation. By combining the adsorption properties of zeolite with the magnetic characteristics of metal oxides, a novel and energy-efficient magnetic adsorbent is created. These particles effectively remove contaminants like antibiotics [13], heavy metals [14], and dyes [15] from a wastewater and are separated via magnetic processes [16]. Currently, zeolite structures are highly valued in catalysis, but their natural forms have limited utility due to several factors: they often contain undesirable impurity phases, their chemical composition can vary widely across different deposits or even within the same deposit, and their properties have not been optimized by nature for catalytic purposes. Therefore, synthetic zeolites are predominantly used for catalytic purposes [17]. Magnetic zeolite NCs are also advanced materials that combine the adsorption capabilities of zeolites with the magnetic properties of NPs, making them highly effective in catalysis. Their key advantages include ease of separation using a magnetic field, applicability in processes such as oxidation, pollutant degradation, chemical synthesis, and biomass conversion, as well as adaptability through functionalization with metal or oxide NPs for specific reactions. These materials are particularly valuable in green chemistry due to their reusability and reduced environmental impact, enhancing sustainability in catalytic applications [18].

Magnetic zeolites have also been utilized to enhance the properties of polymer matrices. Zeolite is a natural microporous material with numerous voids in its structure, enabling it to absorb various polymers and facilitate the preparation of functional composite materials [19]. Furthermore, the biocompatibility of magnetic zeolite NCs also makes them suitable for various medical applications. Several studies have explored the potential application of paramagnetic NP-loaded zeolites as contrast agents for magnetic resonance imaging (MRI) [20]. The key advantages of zeolites compared to other matrices are their complex pore structure and the presence of various active sites on the surface, which also allow for sustained drug release directly from the original matrix without the need for additional surface modification with functional groups, as is typically performed to retain drug molecules within the matrix channels and pores, which is widely used in drug delivery application [21].

Magnetic zeolite NCs can be synthesized and characterized using various materials and methods, depending on the desired properties and use in different applications [22,23,24]. This review article aims to provide a comprehensive overview of the intrinsic properties, classifications, synthesis strategies, and characterization techniques of magnetic zeolite NCs. To offer deeper insight into the material’s evolution, the development of magnetic zeolite NCs over time is included, and particular emphasis is also placed on the broad spectrum of applications associated with magnetic zeolite NCs.

## 2. Magnetic Zeolite Nanocomposites

When MNPs are combined with zeolites in the form of nanostructured systems, they are referred to as magnetic zeolite NCs. These materials combine the properties of zeolites, MNPs, and often other functional components, such as polymers, metals, oxides, etc. [25]. Compared to magnetic zeolites, magnetic zeolite NCs often feature smaller and more uniformly distributed MNPs (e.g., Fe_3_O_4_). As a result, these NPs are also more functional, exhibiting higher adsorption capacity, selectivity, and chemical stability [26,27]. Additional surface modifications enable the use of magnetic zeolite NCs in more specific applications (e.g., functionalization with organic groups or biomolecules). These NCs are synthesized using various methods, either by synthesizing NPs and subsequently combining them with zeolites, or through a combination of chemical, physical, or surface modifications of zeolites and MNPs simultaneously, allowing for more tailored and specific properties [28,29]. The use of polymers or other binders is common, as they stabilize the MNPs. These NCs are also valuable in biomedical fields (e.g., drug delivery, separation, etc.) [28,30] and environmental applications [23,31,32,33], where their higher specific surface area and surface functionalization improve pollutant adsorption. Additionally, in catalytic applications [29,34], magnetic zeolite NCs provide greater catalyst stability and enhanced selectivity in reactions.

As previously mentioned, magnetic zeolite NCs consist of two main components: zeolite and MNPs. Focusing first on the structure of zeolites, these materials are microporous crystalline aluminosilicates with a unique three-dimensional framework that grants them various functional properties, such as adsorption, ion exchange, and catalysis [34,35,36].

The fundamental structure of zeolites comprises SiO_4_ and AlO_4_ tetrahedra, which are interconnected via oxygen bridges to form a three-dimensional network. This framework creates channels and cavities that facilitate the transport and storage of molecules of different sizes. The primary structural unit, or basic building block, of zeolites is the tetrahedron (SiO_4_ or AlO_4_), where silicon (Si^4+^) or aluminum (Al^3+^) occupies the central position, surrounded by four oxygen atoms. The tetrahedra further connect into rings (4-, 5-, 6-, 8-, and 12-membered rings), which form larger structural motifs. These rings create channels, pores, and cavities that define the specific properties of each zeolite [34,37].

The Si/Al ratio is highly significant, as it strongly influences the characteristics of zeolites. A low Si/Al ratio results in a higher negative charge, greater ion exchange capacity, and a more hydrophilic nature (e.g., zeolite A). Conversely, a high Si/Al ratio leads to lower ion exchange capacity, more hydrophobic properties, and improved chemical stability (e.g., ZSM-5) [4,36,37].

Since the aluminum tetrahedron (AlO_4_^−^) carries a negative charge, this must be compensated by cations such as Na^+^, K^+^, Ca^2+^ or Mg^2+^. These exchangeable cations enable the use of zeolites in ion exchange applications, such as water softening and heavy metal removal. Additionally, pore and channel size is one of the key properties of zeolites, as they determine which molecules can enter the structure and be adsorbed. Zeolites are often classified based on pore size, with microporous zeolites (pores < 2 nm)—including most natural and synthetic zeolites—and mesoporous materials (pores 2–50 nm), which are often modified zeolites or composites [4].

Surface functionalization of zeolites is crucial for enhancing their physicochemical properties, such as adsorption capacity, selectivity, hydrophobicity/hydrophilicity, and compatibility with other materials (e.g., MNPs) [38]. Functionalization is performed using chemical or physical methods that modify the surface chemistry of zeolites. When discussing types of surface functionalization, the first approach is ion exchange or cationic modification. Since zeolites contain exchangeable cations (e.g., Na^+^, K^+^, Ca^2+^), these can be replaced with other cations to improve selectivity and adsorption properties. The goal is to enhance selectivity for heavy metals, ammonia, or other contaminants [38].

Acid and alkaline treatment allows for the removal of impurities, adjustment of pore size, and increase in the specific surface area:•Acid treatment (commonly using HCl, HNO_3_, or HF) is employed to remove aluminum and create larger mesopores.•Alkaline treatment with NaOH or KOH promotes the formation of additional pores and increases the specific surface area.

Another type of functionalization is silanization, which involves the attachment of organic silanes (SiR_4_) that introduce functional groups such as –NH_2_, –SH, –COOH, and –OH. This modification enhances hydrophilicity or hydrophobicity and improves interactions with metal NPs.

Additionally, impregnation and coating with metal oxides can enhance adsorption and catalytic properties. Meanwhile, polymeric coatings, such as polyvinyl alcohol (PVA) and polyethylene glycol (PEG), improve the stability and dispersion of NCs in aqueous environments.

Table 1 provides an overview of these functionalization methods, highlighting their main effects and respective application areas [4,38].

MNPs serve as the second key component of magnetic zeolite NCs, enabling easy separation of the material from a solution using an external magnetic field. Their structure and properties are highly dependent on the type of magnetic material, synthesis method, and surface functionalization [39].

MNPs are typically composed of ferromagnetic or superparamagnetic materials, such as iron oxides (Fe_3_O_4_, γ-Fe_2_O_3_), metallic NPs (Fe, Co, Ni), or alloys (FePt, CoPt) [40,41,42]. Their structure is often multilayered, consisting of

A magnetic core, which determines the particle’s magnetic properties.A protective coating, which stabilizes the NPs and prevents oxidation and agglomeration.A functionalized surface, which incorporates chemical groups or coatings that enhance interaction with zeolites and improve adsorption properties.

The most used MNPs in zeolite NCs are magnetite (Fe_3_O_4_) and maghemite (γ-Fe_2_O_3_) due to their biocompatibility, chemical stability, and ease of synthesis [43,44]:•Magnetite (Fe_3_O_4_) exhibits high saturation magnetization and is superparamagnetic at small sizes (<20 nm), meaning it does not retain magnetization after the external field is removed. However, it readily oxidizes into maghemite when exposed to air.•Maghemite (γ-Fe_2_O_3_) is structurally similar to magnetite but is more oxidation-resistant, making it widely used in environmental and biomedical applications.

In addition to iron oxides, metallic NPs are also used in zeolite NCs. These exhibit higher saturation magnetization compared to oxides but suffer from lower stability and higher oxidation rates:•Iron (Fe) NPs have very high magnetization but are prone to oxidation, requiring protective coatings such as SiO_2_ or polymers.•Cobalt (Co) and nickel (Ni) NPs are strong magnets, but their high toxicity and lower chemical stability limit their applications.

A third group includes alloys and composite materials (e.g., FePt, CoPt, FeNi, FeCo) [40,45,46], which exhibit exceptionally high magnetic anisotropy, meaning they retain magnetization even in the absence of an external field (ferromagnetism).

Since MNPs tend to agglomerate and oxidize, their surfaces are often coated with various materials [43,47,48,49]:•Inorganic coatings, such as silica (SiO_2_), carbon layers, or metal oxides (TiO_2_, ZnO, Al_2_O_3_) [43,50,51].•Organic coatings, including polymers (PVA, PEG, chitosan) [48,52].•Silane-based functionalization using (3-Aminopropyl)trietoxysilane (APTES) or tetraethyl ortosilicate (TEOS) [53,54].

MNPs can exhibit different magnetic behaviors, depending on particle size, crystalline structure, and temperature:•Ferromagnetic materials (Fe, Co, Ni) retain magnetization even in the absence of an external magnetic field.•Superparamagnetic materials (Fe_3_O_4_, γ-Fe_2_O_3_ < 20 nm) exhibit magnetization only when an external magnetic field is applied, making them ideal for NCs, as they do not agglomerate and allow for efficient separation without residual magnetization [55,56].

The interactions between zeolites, MNPs, and other functional components play a crucial role in determining the properties of magnetic zeolite NCs [57]. These interactions can be classified into physical and chemical interactions, where surface bonds, electrostatic forces, and functional groups are key factors [38,58].

Physical adsorption involves non-covalent interactions, which are generally weaker but still essential for NC stability and adsorption capacity. These interactions include the following:•Van der Waals forces arise due to temporary dipoles between molecules and the NC surface. These forces contribute to the adsorption of non-polar molecules and certain pollutants, such as organic contaminants and oils.•Electrostatic interactions [59], where zeolites, due to their negatively charged framework resulting from exchangeable cations (Na^+^, K^+^, Ca^2+^), attract positively charged ions and cationic compounds. On the other hand, MNPs (Fe_3_O_4_, γ-Fe_2_O_3_) possess a surface charge that can change with pH, affecting interactions with other components. These electrostatic interactions can be enhanced by adjusting pH or ionic strength of the solution.•Hydrogen bonding is crucial for interactions between hydroxyl (-OH) groups on the surfaces of zeolites, MNPs, or functionalized materials (e.g., silanized surfaces). Hydrogen bonds facilitate the adsorption of polar molecules, including heavy metals and certain organic compounds [59,60].

Chemical adsorption involves strong interactions, where the formation of covalent bonds or coordination complexes occurs:•Covalent bonds: These play a crucial role in surface functionalization. A key example is the silanization of MNPs (Fe_3_O_4_) with APTES, which introduces -NH_2_ groups [61]. These groups can further form covalent bonds with contaminants or catalytic centers, enhancing the nanocomposite’s reactivity.•Complexation with metal ions [31]: Zeolites and MNPs can form coordination complexes with metal ions, improving their adsorption capacity. For instance, Fe_3_O_4_ NPs functionalized with carboxyl (-COOH) or amino (-NH_2_) groups can selectively complex Cu^2+^ and Pb^2+^ ions, increasing their removal efficiency.•Ion exchange: Zeolites facilitate the exchange of cations within their structure, enhancing the removal of heavy metals. A notable example is magnetic zeolite NCs functionalized with amino groups (-NH_2_), which exhibit improved adsorption of heavy metal ions (Pb^2+^, Hg^2+^) due to their ability to bind metal ions through complexation [38,57].

In addition to zeolites and MNPs, other components can be incorporated into magnetic zeolite NCs to enhance their adsorption capacity, stability, catalytic properties, or selectivity. Among these, metal oxides and metallic NPs play a significant role in improving functional properties. For instance, titanium dioxide (TiO_2_) [31] is introduced due to its photocatalytic activity, enabling the degradation of organic pollutants under UV light. Zinc oxide (ZnO) [31] is added to enhance the antibacterial properties of the NCs, while silver NPs provide strong antimicrobial and antibacterial effects [62,63]. Gold NPs contribute to increased selectivity in biomolecule adsorption [64], whereas copper NPs facilitate catalytic reactions for the decomposition of organic pollutants [65].

In addition to metal-based additives, organic polymers and biopolymers are often incorporated to improve adsorption performance and mechanical stability. Polymers such as PVA and PEG contribute to better dispersion and stabilization of the NC, while biopolymers such as chitosan, alginate, and cellulose derivatives enhance adsorption efficiency, particularly for dyes and pharmaceutical compounds in water [38]. A notable example is chitosan-functionalized magnetic zeolite NCs, which have demonstrated high efficiency in the removal of lead (Pb^2+^) and hexavalent chromium (Cr^6+^) from wastewater [23].

Surface functionalization also plays a crucial role in tailoring the properties of magnetic zeolite NCs. Silanization using agents such as APTES and TEOS facilitates the binding of organic molecules and metal ions to the surface [34]. Sulfonation, which introduces –SO_3_H functional groups, improves the adsorption of heavy metals and enhances catalytic activity, while carboxylation (–COOH groups) increases the affinity of the NC for cations such as Pb^2+^ and Cd^2+^ [66,67,68].

The interactions between the various components in magnetic zeolite NCs are governed by both physical and chemical adsorption mechanisms, with electrostatic forces, metal ion complexation, and ion exchange playing key roles. The inclusion of additional components such as TiO_2_, ZnO, metallic NPs, polymers, and biopolymers further enhances the material’s functionality, making it suitable for a wide range of environmental, catalytic, and biomedical applications.

As previously described in this chapter, magnetic zeolite NCs consist of various combinations of zeolites and MNPs, which differ in composition, shape, size, and dispersion. Loiola and others [29] based on an extensive review of research published over the past 25 years, categorize these materials into five distinct types (I, II, III, IV, and V), as presented in Figure 1. This classification is based on the arrangement of magnetic particles relative to the zeolite crystals.

Type I refers to magnetic zeolite composites where zeolite crystals are formed in the presence of MNPs, leading to the encapsulation of the NPs within the zeolite structure. While this process may appear straightforward, it comes with several challenges. The inclusion of multiple phases in the reaction mixture for zeolite crystallization can result in the formation of unintended phases. Additionally, there is a limit to how many magnetic particles can be incorporated into the zeolite crystals without disrupting their integrity. At the same time, a sufficient quantity of magnetic particles is necessary to ensure the composite exhibits effective magnetic properties.

The preparation of type II zeolite magnetic composites predominantly involves the impregnation of MNPs onto the surface of zeolites. This approach leverages the simplicity of the related methods and the wide variety of available zeolite and magnetic materials. In essence, magnetic particles can be deposited on zeolite surfaces by bringing them into contact in an aqueous medium, often with the aid of energy inputs such as heating, ultrasound, or microwaves. However, a key challenge arises when Fe_3_O_4_ particles are dispersed on the zeolite surface without a protective layer, as they are prone to oxidation, leading to a loss of magnetic properties.

Type III composites encompass magnetic zeolite composites composed of single magnetic particles coated with zeolite layers. To facilitate the growth of zeolite crystals around the magnetic core, an intermediate layer, such as silica, is often necessary. Silica serves as a non-magnetic and relatively inert shell, preserving the magnetic properties of the particles by preventing oxidation. Simultaneously, it aids in stabilizing the zeolitic crystal formation on the magnetic core.

In some instances of magnetic zeolite NPs, the particle sizes of both the zeolites and the magnetic compounds are observed to be below 100 nm. Materials with these characteristics, classified here as type IV, are particularly noteworthy because their small particle size enhances interactions between particles, thereby reducing the likelihood of agglomeration. Additionally, a minimal impact on the surface area is anticipated.

Type V magnetic zeolite composites offer promising solutions to challenges such as particle agglomeration, oxidation of magnetic components, and leaching losses. Their core concept involves embedding the magnetic composites within a polymeric matrix, which provides an impermeable barrier that shields the NPs from oxygen exposure. This approach not only protects the magnetic components but also imparts additional properties such as enhanced mechanical, chemical, and electrical stability. While these composites are primarily composed of a polymer as the third component, other materials are sometimes incorporated as well [29].

Magnetic zeolite NPs combine the adsorption properties of zeolites with the magnetic properties of NPs, making them highly effective for environmental, industrial, and biomedical applications [16,28,29]. Their key feature is their superparamagnetic behavior, which allows for easy separation and reuse of the NCs after pollutant adsorption [29,33]. This property ensures that the material does not retain magnetization once the external magnetic field is removed, enabling rapid separation from aqueous systems without the need for filtration, which can be time-consuming and inefficient. The responsiveness of these NCs to external magnetic fields also allows for targeted applications, such as drug delivery or selective pollutant removal from contaminated water [28,69]. However, maintaining the stability of the magnetic properties is crucial for long-term applications, requiring coatings such as SiO_2_, polymers, or carbon-based layers to prevent oxidation and ensure stable dispersion in water.

The exceptional adsorption capacity of magnetic zeolite NCs is primarily due to the porous structure of zeolites, which provides a high specific surface area. MNPs further enhance the accessibility of active sites within zeolites. Additionally, the presence of exchangeable cations (Na^+^, K^+^, Ca^2+^) in zeolites facilitates the selective removal of heavy metals such as Pb^2+^, Cd^2+^, and Hg^2+^ [70]. Surface functionalization, including silanization, sulfonation, and carboxylation, further improves the selective binding of specific contaminants, tailoring the material for targeted adsorption.

Another critical property of these NCs is their thermal stability. Zeolites can withstand high temperatures (>700 °C), making them suitable for catalytic reactions [71,72]. However, MNPs, particularly Fe_3_O_4_, begin to oxidize at temperatures above 200–300 °C, limiting their application in high-temperature processes [39]. Chemical resistance is another key factor, as zeolites exhibit excellent stability in acidic and basic environments, while MNPs are more susceptible to oxidation in acidic conditions, where Fe_3_O_4_ can convert into Fe^3+^. The addition of protective coatings, such as SiO_2_ or polymers, enhances chemical stability and prevents degradation [37,73].

To improve mechanical stability and prevent brittleness, polymers such as PVA and chitosan are often incorporated, reinforcing the structural integrity of the NC [74]. Additionally, the catalytic properties of these materials can be significantly enhanced by incorporating TiO_2_ or ZnO, which facilitates the degradation of organic pollutants under UV or visible light [31]. When doped with metallic NPs (Ag, Au, Pt, Cu), the NCs act as catalysts for oxidation and reduction reactions, and in some cases, also exhibit antimicrobial properties, making them useful in disinfection applications [63,64,65,69].

An essential aspect of magnetic zeolite NPs is their non-toxicity and biocompatibility [29]. Zeolites are naturally environmentally friendly and non-toxic, whereas MNPs can induce oxidative stress in cells. However, this effect can be mitigated through surface coatings such as SiO_2_, polymers, or chitosan, making them safer for biomedical applications [75]. Furthermore, these NPs offer the advantage of reusability, as they can be magnetically separated and regenerated, reducing operational costs and waste production. Adsorbed contaminants can be desorbed by altering pH or temperature, allowing for repeated use of the material.

The introduction of conductive materials such as graphene and carbon nanostructures imparts electrical conductivity to these NPs, enabling their application in sensors and electrocatalysis [57]. Meanwhile, doping with Au or Ag NPs enhances plasmonic resonance, giving them optical properties that are valuable for spectroscopic applications [33,57].

Overall, magnetic zeolite NPs exhibit outstanding adsorption, magnetic, catalytic, and mechanical properties, making them highly versatile for a broad range of applications. Their selective adsorption capabilities, chemical stability, regenerability, and biocompatibility make them particularly promising for environmental remediation, catalysis, and biomedical technologies.

## 3. The Development of Magnetic Zeolite Nanocomposites over Time

Research on magnetic zeolite NCs has evolved over several decades, with scientists progressively discovering and enhancing their properties and applications. Below is an overview of key milestones in the development of these materials.

In the 1990s, extensive research was conducted on the synthesis, characterization, and stabilization of MNPs, particularly magnetite (Fe_3_O_4_), due to its unique superparamagnetic properties and potential applications in separation technologies, catalysis, and biomedicine [39,55,76]. Researchers investigated various synthetic routes, including co-precipitation, thermal decomposition, and hydrothermal methods, aiming to optimize particle size, crystallinity, and magnetic behavior. One of the key challenges was preventing aggregation and oxidation of Fe_3_O_4_ NPs, leading to the development of surface coatings such as silica (SiO_2_), polymers, and surfactants to enhance their colloidal stability and reusability [77,78].

Simultaneously, zeolites attracted significant attention due to their well-defined microporous structures, high surface area, and ion exchange properties, making them highly effective adsorbents and catalysts. Studies in this period focused on modifying the physicochemical properties of zeolites to improve their selectivity for specific contaminants, such as heavy metal ions and organic pollutants. Researchers explored ion exchange mechanisms in zeolites, particularly for Pb^2+^, Cd^2+^, and Hg^2+^ removal, and investigated their catalytic behavior in acid–base and redox reactions [79,80,81,82,83,84,85].

The early 2000s marked the emergence of magnetic zeolite NCs as a novel class of hybrid materials, integrating the high adsorption efficiency of zeolites with the magnetic properties of Fe_3_O_4_ NPs. Scientists recognized the potential of these composites for rapid separation, easy recovery, and reusability in environmental remediation. Initial studies focused on optimizing the synthesis of magnetic zeolite composites, including direct precipitation of Fe_3_O_4_ within zeolite frameworks and impregnation methods to achieve uniform distribution of NPs [86,87,88,89,90]. Researchers also investigated the physicochemical interactions between MNPs and zeolite surfaces, such as electrostatic interactions, ion exchange, and hydrogen bonding, which influence the stability and adsorption performance of the composites [91,92,93].

Early applications demonstrated the effectiveness of magnetic zeolite NCs in removing heavy metals from aqueous solutions, with studies reporting high adsorption capacities and fast separation kinetics under applied magnetic fields [94,95]. These findings paved the way for further advancements in functionalizing magnetic zeolite surfaces to enhance selectivity, stability, and catalytic efficiency, marking a significant step towards practical applications in environmental and industrial settings.

By the mid-2000s, research efforts had shifted towards refining synthesis techniques and improving the physicochemical properties of magnetic zeolite NCs to enhance their performance in environmental and catalytic applications. One of the major challenges was achieving a homogeneous dispersion of Fe_3_O_4_ NPs while preventing agglomeration, which could negatively affect the adsorption efficiency and magnetic response of the composites. To address this, researchers investigated surface functionalization strategies, such as silanization, polymer coating, and carbon-based modifications (graphene, carbon nanotubes), to improve the stability and dispersibility of MNPs within zeolites [96,97,98].

A key focus during this period was understanding the impact of these surface modifications on the adsorption capacity, selectivity, and regeneration potential of NCs. Studies demonstrated that functionalized magnetic zeolite NCs exhibited improved adsorption affinity for specific contaminants, including heavy metal ions (Pb^2+^, Cd^2+^, Cr^6+^) and organic pollutants (dyes, pharmaceuticals), due to enhanced surface interactions, increased active sites, and improved mass transfer efficiency [25,99,100,101,102,103]. Additionally, efforts were made to enhance the thermal and chemical stability of these materials to extend their applicability to diverse environmental conditions.

By the late 2000s, research had transitioned towards real-world applications of magnetic zeolite NCs, with a strong emphasis on wastewater treatment, catalytic degradation, and separation technologies. Scientists explored the potential of these NCs for the rapid and efficient removal of heavy metals and persistent organic pollutants from aqueous solutions under magnetic field-assisted separation [26,104,105,106]. Investigations into their catalytic properties gained traction, particularly in the context of Fenton-like reactions and photocatalysis, where nanocomposites doped with transition metals (Fe, Cu, Mn) or semiconductor oxides (TiO_2_, ZnO) demonstrated enhanced degradation of organic contaminants [107,108].

An important aspect of late-2000s research was the regeneration and reusability of magnetic zeolite NCs, crucial for their practical implementation. Studies assessed their stability under varying environmental conditions, such as different pH levels, ionic strengths, and long-term operational cycles. Results indicated that certain functionalized NCs could be effectively regenerated through desorption techniques involving pH adjustments or thermal treatments, retaining a high percentage of their adsorption efficiency over multiple cycles [7,109].

During the 2010s, the scope of research expanded significantly, particularly towards biomedical applications. Magnetic zeolite NCs were explored as potential drug delivery systems, leveraging their porous structure for drug encapsulation and controlled release, while their magnetic properties enabled targeted delivery under an external magnetic field [101,110,111,112,113]. Concurrently, their use in magnetic separation of biomolecules, such as proteins and nucleic acids, was investigated due to their high surface area, selective adsorption capacity, and ease of separation via magnetic fields [32,69,114].

By 2020, research had become increasingly focused on fine-tuning surface functionalization strategies to enhance the selectivity of magnetic zeolite NCs for specific contaminants. Targeted modifications, such as sulfonation, amination, and thiol-functionalization, were employed to improve the affinity of these materials for heavy metals (As^3+^, Hg^2+^) and organic pollutants (antibiotics, endocrine-disrupting compounds) [115,116]. These advancements marked a shift towards application-specific NC design, optimizing their performance for diverse environmental and biomedical applications.

In the early 2020s, research on magnetic zeolite NCs experienced significant advancements, particularly in the development of new synthesis techniques and the expansion of their applications. One of the key methods that gained prominence was microwave-assisted synthesis, which enables a faster and more uniform formation of NCs. For example, Haghdoust et al. [117] developed a magnetic biochar@ZIF-67 NC using microwave synthesis, demonstrating high efficiency in removing anionic dyes from contaminated water. During this period, there was also a growing interest in using magnetic zeolite NCs for the removal of micro- and nanoplastics from the environment. Pasanen et al. [118] synthesized a magnetic ZIF-8 NC, enabling the simultaneous and efficient removal of polystyrene (PS) microspheres and endocrine disruptors (EDs) from aqueous solutions. By 2024, ongoing research continues to focus on optimizing synthesis methods to enhance the efficiency and stability of NCs and expand their applicability in various environmental applications [28,29,33,60,119]. A particular emphasis is placed on the development of materials capable of efficiently removing micro- and nanoplastics from water systems, which is crucial for mitigating environmental pollution caused by these persistent contaminants. Vohl et al. [120], in their review, highlighted the potential of MNPs for removing micro- and nanoplastics from water sources, opening new possibilities for the use of magnetic zeolite NCs in environmental remediation.

This timeline illustrates the evolution of research on magnetic zeolite NCs, from early studies of their fundamental properties to modern applications in environmental remediation and biomedicine.

Figure 2 illustrates the chronological development of research on magnetic zeolite NCs, highlighting key milestones from the synthesis and characterization of fundamental structures in the 2000s to their advanced applications in environmental remediation, pollutant removal, and biomedical fields in the 2020s.

## 4. Synthesis Methods of Magnetic Zeolite Nanocomposites

Magnetic zeolite NCs are hybrid materials in which MNPs are embedded in a zeolite structure or deposited on their surface. As a result, the material acquires magnetic properties while retaining the beneficial characteristics of zeolites, such as porosity, high specific surface area, chemical stability, and adsorption capacity [11,121]. Most incorporated in zeolite structure are MNPs made of elemental metals or their oxides, usually including magnetite (Fe_3_O_4_), maghemite (Fe_2_O_3_), cobalt ferrite (CoFe_2_O_4_), and nickel ferrite (NiFe_2_O_4_). These composites combine the benefits of all the materials used; zeolites gain magnetism from MNPs, with minimal structural change [122]. In the case of composites consisting of MNPs and zeolites, the latter acquire magnetic properties. However, the modification is expected to be moderate enough to avoid significant alterations to the zeolite structure while being strong enough to impart magnetic properties [123].

The synthesis of magnetic zeolite composites combines techniques for synthesizing MNPs and zeolites to create a hybrid material that integrates the advantages of both components.

Various methods are used to prepare MNPs (Figure 3), including co-precipitation, microemulsion, thermal decomposition, hydrothermal, sonochemical, microwave-assisted synthesis, sol–gel synthesis, and electrochemical synthesis. MNPs can also be synthesized by other, less commonly used methods, such as laser pyrolysis techniques, chemical vapor deposition, combustion, carbon arc, laser pyrolysis, and more [124].

Table 2 presents the most commonly used synthesis methods for MNPs, highlighting their advantages and limitations.

Magnetic iron oxide NPs have a large surface-to-volume ratio and consequently possess high surface energy. Therefore, they tend to aggregate to minimize surface energy. Moreover, the bare iron oxide NPs have high chemical activity, and are easily oxidized in air, generally resulting in the loss of magnetism and dispersibility. Therefore, it is important to provide proper surface coating and improve some effective protective approaches to retain their stability [124]. Developing effective strategies to enhance the chemical stability of MNPs is essential. One of the simplest approaches is to shield them with an impenetrable layer, preventing oxygen from reaching their surface. In many cases, stabilization and protection of the particles are closely interconnected. Coating strategies can generally be classified into two main categories: organic coatings, which include surfactants and polymers, or inorganic coatings [138], which encompass materials such as silica, carbon, precious metals (e.g., Ag, Au), or oxides [139]. The choice of coating material for MNPs depends on their intended application. In general, surface coating can influence the magnetic behavior of MNPs. Therefore, studying the magnetic properties of MNPs with different coatings is crucial for achieving coatings that vary in composition but maintain similar magnetic characteristics [140].

**Surfactant- and polymer-coated** MNPs have attracted considerable interest in applications in both biomedicine and technical fields. Their synthesis generally follows a two-step process: first, the formation of the magnetic core, followed by the application of a polymer coating [141]. Surfactants or polymers can be chemically anchored or physically adsorbed onto MNPs, forming a single or double layer that stabilizes them in suspension through steric repulsion. Functional polymers, such as those with carboxylic acid, phosphate, or sulfate groups, can bind to MNPs. Common coating polymers include poly(pyrrole), poly(aniline), polyesters like poly(lactic acid) and poly(glycolic acid), and their copolymers. Surface-modified MNPs with biocompatible polymers are widely studied for drug targeting and as MRI contrast agents [39,142,143]. **Precious metals** can be deposited on MNPs using methods like redox transmetalation or hydroxylamine seeding to prevent oxidation. Gold coatings are particularly useful, as their surface can be functionalized with thiol groups, enabling the attachment of ligands for catalytic and optical applications [39,144].

A **silica shell** not only protects the magnetic core but also prevents direct contact with surface-linked agents, minimizing unwanted interactions. Silica-coated MNPs are hydrophilic and easily functionalized with various groups, enabling applications in biolabeling, drug targeting, and drug delivery. However, silica is unstable in basic conditions and may contain pores that allow oxygen or other species to diffuse through [39]. Although most studies so far have concentrated on developing polymer or silica protective coatings, **carbon-protected** MNPs are gaining increasing attention. This is because carbon-based materials offer several advantages over polymers or silica, including significantly higher chemical and thermal stability, as well as improved biocompatibility. The well-developed graphitic carbon layers form an effective barrier against oxidation and acid erosion. This suggests that carbon-coated MNPs can be synthesized to exhibit high thermal stability, along with strong resistance to oxidation and acid leaching, making them suitable for various applications [39,145].

Several different synthesis methods are available for preparing magnetic zeolite NCs, allowing control over their structure, properties, and efficiency in targeted applications:•**Impregnation method**: The zeolite structure is immersed in a solution containing precursors of MNPs, usually iron salts such as FeCl_2_ or FeCl_3_. This is followed by a reduction or precipitation process, where iron ions are converted into MNPs, such as magnetite or maghemite [29]. This type of reaction was carried out under mild conditions using low-energy input and inexpensive, non-toxic materials, resulting in inert residues and avoiding hazardous solvents. These features highlight the method’s alignment with green chemistry principles, representing its key advantages.•A rapid, environmentally friendly impregnation method was also used to prepare MNPs on sodium/potassium zeolite surfaces. Using ferric and ferrous chloride with sodium hydroxide, the zeolite/Fe_3_O_4_ NCs was formed in aqueous suspension under ambient conditions, following green chemistry principles [146]. Pescarmona et al. [147] developed a method for the easy separation of heterogeneous catalysts from liquid reaction mixtures in high-throughput experiments (HTEs) using magnetically modified zeolites. Specifically, the zeolites were impregnated with an aqueous solution of an iron precursor, and after reduction in hydrogen, ferromagnetic iron oxide NPs formed on the surface of the zeolites. Vajglova et al. [148] synthesized a series of mono- and bimetallic catalysts by impregnating H–Y-5.1 zeolite with iron and nickel nitrates. These catalysts were prepared with varying Fe/Ni ratios and subsequently calcined.•**In situ synthesis**: MNPs are synthesized directly within the porous zeolite structure during the zeolite formation process [149]. Zhang et al. [150] introduced a novel method for synthesizing NaP zeolite adsorbents doped with transition metals (M-NaP) utilizing fly ash as a raw material. The process involves extracting sodium silicate (Na_2_SiO_3_) and sodium aluminate (NaAlO_2_) from fly ash through activation and staged treatment. The in situ synthesis is combined with an organic complexation method to incorporate transition metals such as Co, Ni, Fe, and Ti into the zeolite framework. Nasir et al. [149] presented a straightforward method for the in situ synthesis of magnetic Fe@Si/zeolite Na composites, in which Fe_3_O_4_ NPs are incorporated into the zeolite structure during the synthesis process. Natural materials were used, and the Fe_3_O_4_ MNPs were prepared via a co-precipitation method, forming core–shell structures with zeolite.•**Co-precipitation**: This method involves the simultaneous precipitation of MNPs and zeolite precursors in a solution, leading to the concurrent formation of both components. It is a simple process that allows for the simultaneous synthesis of NPs and zeolites [29]. Nabiyouni et al. [98] synthesized the Fe_3_O_4_ NPs and their incorporation into zeolite-Y matrices using a chemical precipitation method. Structural and morphological analyses confirmed the successful formation of NCs, with Fe_3_O_4_ particles uniformly distributed within the zeolite framework.•**Microwave-assisted method**: Microwave-assisted methods provide a fast and effective route for synthesizing magnetic zeolite NCs, enabling shorter reaction times and often producing materials with improved uniformity and fewer structural defects compared to traditional hydrothermal techniques [151]. Piri et al. [15] presents the development of a magnetic zeolite–hydroxyapatite (MZeo-HAP) NC synthesized via a microwave-assisted method. The process involves reinforcing magnetic hydroxyapatite with zeolite to create an efficient adsorbent.•**Mechanical synthesis (milling)**: Zeolite and MNPs are mechanically mixed using a milling device, such as a ball mill. The milling process ensures the uniform dispersion of MNPs within the zeolite powder. The method begins by reducing zeolite particles to the nano- or microscale, after which iron oxide nanocrystals are synthesized in their presence. This approach effectively minimizes the agglomeration of magnetite NPs and promotes their uniform integration into the zeolite matrix [152]. Murrieta-Rico et al. [153] explores a solvent-free mechanochemical approach to synthesize iron-modified MFI zeolites. By grinding ammonium-form MFI zeolite with iron(III) chloride, researchers achieved the incorporation of iron into the zeolite framework.•**Hydrothermal synthesis**: MNPs are synthesized within the zeolite structure under high temperature and pressure conditions, typically in an autoclave. Aboelfetoh et al. [154] presented a simple one-step hydrothermal synthesis method for a magnetic and porous zeolite/SnFe_2_O_4_ NC designed for the removal of both cationic and anionic dyes from wastewater. Characterization confirmed the successful integration of SnFe_2_O_4_ NPs into the zeolite structure, resulting in a material with high surface area, strong magnetic properties, and efficient adsorption capabilities.•**Pyrolysis**: This process involves heating the precursors of MNPs in the presence of zeolites using a flame or a hot gas stream, leading to the formation of MNPs [155]. Gao et al. [156] explored an innovative method for synthesizing magnetic zeolite composites by utilizing pyrolysis products derived from waste printed circuit boards (WPCBs). The researchers employed the residual heat and carbon-rich gases from the pyrolysis of WPCBs to facilitate the formation of carbon fibers on waste zeolites, resulting in magnetic zeolites coated with carbon fibers.

Each method for synthesizing magnetic zeolite composites has its advantages and drawbacks. The choice of method depends on the desired properties of the final composite, including particle size, distribution of MNPs, cost, and process complexity [29].

MNPs and zeolites can be, as already mentioned above, functionalized with various functional groups; both possess external surfaces that can be easily modified with organic functional groups. This modification allows for the tailoring of their adsorption properties to suit specific applications [157]. Introducing functional groups such as amines, carboxyls, or silanes improves the overall performance of the nanocomposite in areas like catalysis, environmental remediation, and drug delivery [158].

## 5. Characterization of Magnetic Zeolite Nanocomposites

A thorough characterization of magnetic zeolite NCs using various analytical techniques is crucial for a complete understanding of their properties. These methods enable the determination of structural, morphological, chemical, magnetic, and surface properties of the materials and are essential for optimizing their synthesis and applications.

### 5.1. X-Ray Powder Diffraction (XRD)

XRD is used to identify the crystalline phases in NCs. In the case of magnetic zeolite NCs, the analysis reveals the presence of characteristic diffraction peaks of MNP and zeolite structures, confirming the successful incorporation of MNPs into the zeolite matrix [159]. Additionally, it allows for the evaluation of crystallinity and crystallite size. The primary purpose of XRD is to determine the crystalline structure, phase composition, and crystallite size in the NC. The technique is based on the scattering of X-rays based on the crystal structure of the material, producing diffraction patterns characteristic of specific crystalline phases. As a result, an XRD analysis of magnetic zeolite NCs provides information on the presence of zeolite and magnetic phases, confirms the successful incorporation of MNPs into the zeolite matrix, and allows for the estimation of crystallite size using the Scherrer equation. Moreover, it can also detect changes in crystallinity due to functionalization or the addition of other components (e.g., TiO_2_, Ag, Au). However, it should be noted that XRD has certain limitations. For example, it cannot detect amorphous phases or low-concentration crystalline components, which may lead to incomplete phase identification. In addition, peak broadening in nanoscale materials can make the accurate determination of crystallite size and phase separation more challenging.

### 5.2. Fourier Transform Infrared Spectroscopy (FTIR)

FTIR enables the analysis of functional groups in the NC. For magnetic zeolite NCs, characteristic absorption bands appear at wavenumbers corresponding to Fe–O vibrations of magnetite (~580 cm^−1^) and Si–O and Al–O bonds in zeolites (~1000 cm^−1^). This technique is particularly useful for confirming the successful functionalization of the surface with various organic or inorganic groups (e.g., -NH_2_, -COOH, -SO_3_H) [28]. The purpose of FTIR is to identify functional groups on the surface of the NC by analyzing the absorption of infrared light at different wavenumbers, allowing for the recognition of chemical bonds and specific functional groups. As mentioned, FTIR analysis monitors the presence of Si–O–Si and Si–O–Al vibrations characteristic of zeolites, confirms the presence of MNPs (Fe–O vibrations at ~580 cm^−1^), identifies functional groups (-OH, -COOH, -NH_2_) important for contaminant adsorption, and verifies the success of functionalization (e.g., silanization, sulfonation, carboxylation). Despite its usefulness, FTIR also has limitations. The technique provides primarily qualitative or semi-quantitative information and may not detect functional groups present in very low concentrations. Additionally, overlapping absorption bands can hinder the clear identification of specific bonds, especially in complex hybrid materials such as magnetic zeolite NCs.

### 5.3. Thermogravimetric Analysis (TGA)

TGA is used to study the thermal stability of NCs and determine the content of magnetic and zeolite components. When heating the NC, mass loss can be identified due to the removal of adsorbed water, decomposition of functional groups, or oxidation of MNPs. The main goal is to analyze thermal stability and determine the content of individual components by heating the material under controlled conditions while monitoring mass loss as a function of temperature [35]. The results of TGA provide insights into the stability of the NC at high temperatures, water content, organic functionalization groups, and MNPs, as well as the decomposition of polymeric components (e.g., chitosan, PVA) at specific temperatures. However, TGA has limitations, particularly in distinguishing overlapping decomposition events, which can make it difficult to attribute mass loss to specific components in complex NCs. In addition, TGA does not provide direct structural or compositional information, and complementary techniques (e.g., FTIR, elemental analysis) are often needed to fully interpret the thermal behavior.

### 5.4. Transmission and Scanning Electron Microscopy (TEM, SEM)

TEM and SEM provide insight into the morphology, particle size, and distribution within the NC. SEM reveals the surface topography and agglomeration of NPs, while TEM enables a more detailed examination of the internal structure of the NC and confirms the incorporation of MNPs into the zeolite framework. The primary purpose of these techniques is to analyze the morphology, particle size, and component distribution [29]. SEM uses an electron beam to capture surface images, whereas TEM provides a detailed view of the internal structure of the material. The results allow for the determination of the size and shape of magnetic and zeolite particles, the homogeneity of MNP distribution within the zeolite matrix, and the confirmation of coatings (SiO_2_, polymers) around MNPs. Despite their high resolution, these techniques have limitations. SEM provides only surface information and requires conductive coatings for non-conductive samples, which can alter the native structure. TEM, although powerful, involves complex sample preparation and operates under vacuum, potentially affecting sensitive samples. Furthermore, the limited field of view may not fully represent the sample’s overall morphology or particle distribution.

### 5.5. Nitrogen Adsorption/Desorption (BET Analysis)

BET analysis is commonly used to determine the specific surface area of nanocomposites by measuring Nitrogen Adsorption/Desorption isotherms. For magnetic zeolite NCs, BET provides insights into the surface area, pore volume, and pore size distribution, particularly after functionalization or the incorporation of MNPs [29,119]. However, it is important to note that BET analysis is not ideally suited for microporous materials such as zeolites, as it may underestimate the true surface area due to the limitations of the BET model in describing micropore filling. In such cases, additional techniques like the t-plot method, αs-method, or density functional theory (DFT) modeling are more appropriate and provide a better representation of the micropore structure. Despite these limitations, BET remains a widely used technique for comparative purposes, especially when evaluating changes in porosity due to synthesis modifications.

### 5.6. Vibrating Sample Magnetometry (VSM)

VSM is used to analyze the magnetic properties of NCs, including saturation magnetization, coercivity, and remanence. Magnetic zeolite NCs typically exhibit superparamagnetic behavior, meaning that the material retains magnetization in the presence of an external magnetic field but does not show residual magnetization after the field is removed, allowing for easy separation and reuse. The primary goal is to determine the magnetic properties of the NC (saturation magnetization, coercivity, remanence), with VSM measuring the response of the NC to an external magnetic field and determining key magnetic parameters [29]. The results provide insights into the saturation magnetization (*M*_s_), the influence of MNPs on overall magnetization, superparamagnetism (absence of remanence, important for easy separation and reuse), and the stability of magnetic properties after functionalization or under different conditions. However, it should be noted that VSM provides bulk magnetic data and does not distinguish between magnetic contributions from different phases or particles. Additionally, magnetic measurements can be influenced by factors such as particle agglomeration, surface effects, or matrix interactions, which may complicate interpretation. Complementary techniques such as Mössbauer spectroscopy or SQUID magnetometry may be required for more detailed magnetic characterization.

### 5.7. Zeta Potential (ZP)

ZP is used to determine the surface charge of NCs in different pH environments. This is crucial for understanding the stability of suspensions in water and the adsorption capacity of materials for various contaminants. Positively or negatively charged surfaces influence the affinity of NCs for specific ions and molecules. Using dynamic light scattering (DLS), the surface charge and dispersion stability of NCs in water can be determined, with ZP being measured by tracking the electrophoretic mobility of particles in an electric field [35]. The results provide information on surface charge at different pH values, which is important for interactions with ions and adsorption selectivity, as well as the stability of NC dispersions in water (higher ZP values indicate better stability). Nevertheless, ZP measurements can be influenced by factors such as ionic strength, the presence of surfactants, and particle aggregation, which may lead to variability in results. Additionally, ZP does not provide direct information about the specific functional groups on the surface, so it is often used in combination with FTIR or XPS for a more comprehensive surface characterization.

### 5.8. Inductively Coupled Plasma Optical Emission Spectroscopy (ICP-OES)

ICP-OES is used for the quantitative analysis of metal ion concentrations before and after adsorption on the NC. This technique enables the evaluation of the efficiency of heavy metal removal, such as Pb^2+^, Cd^2+^, and Hg^2+^, and determines the adsorption capacity of the material [160]. However, sample preparation steps—such as acid digestion or dilution—can introduce variability or contamination if not carefully controlled. Moreover, the method does not provide information about the binding mechanism or the speciation of adsorbed metals, so it is often used in conjunction with other characterization techniques such as FTIR or XPS to obtain a more complete understanding of adsorption processes.

### 5.9. X-Ray Photoelectron Spectroscopy (XPS)

XPS determines the elemental composition and oxidation states on the surface of the NC. XPS analyzes electrons emitted by atoms under the influence of X-ray radiation [29,153]. The results provide information on the presence of Fe, Si, Al, O, and other elements on the surface, identify the oxidation states of iron (Fe^2+^/Fe^3+^) in MNPs, and confirm the presence of functional groups and metal dopants (Ag, Au, Cu). Despite its high surface sensitivity and ability to provide oxidation state information, XPS is limited to the analysis of only the top few nanometers of the material’s surface and may not fully reflect the bulk composition. Additionally, sample charging (especially in non-conductive materials like zeolites) and the need for ultra-high vacuum conditions can influence the accuracy of the results. Therefore, careful calibration and complementary techniques are often necessary to obtain a complete interpretation.

The characterization of magnetic zeolite NCs is essential for understanding their physicochemical properties, adsorption capacity, and application potential. By employing a combination of various analytical techniques, synthesis procedures can be optimized, functional properties improved, and the effective use of these materials ensured in environmental and biomedical applications. Figure 4 provides a schematic overview of the characterization techniques discussed in this section. It highlights the most relevant parameters obtained from each method, such as particle size and structure, morphology, surface functional groups, thermal stability, porosity, crystallinity, and magnetic properties, which are critical for the comprehensive evaluation of magnetic zeolite NCs.

## 6. Applications of Magnetic Zeolite Nanocomposites

Magnetic zeolite NCs exhibit excellent efficiency in various fields. Due to their multifunctionality, they have a wide range of potential applications (Figure 5), such as catalysis, bioreactors, sorbents, ion exchange, contrast agents for MRI, and drug delivery. One of the most promising and widespread applications is their use in water purification, as they offer a cost-effective and environmentally friendly solution compared to other approaches, which is also reflected in the growing number of scientific publications on this topic [29].

Organic and inorganic contaminants, such as pesticides, antibiotics, dyes, and heavy metals, pose risks to human health and the environment, necessitating their removal. A growing variety of pollutants, including priority chemicals, heavy metals, and other contaminants, enter water bodies through industrial discharges and agricultural runoff containing pesticides and fertilizers. Poor management of these pollutants leads to waterborne diseases and other health risks. Moreover, prolonged exposure to contaminated water can cause serious health issues, such as cancer, developmental disorders and reproductive problems [161]. Various methods have been utilized for removing these contaminants in water and wastewater treatment, including ion exchange, sedimentation, membrane filtration, chemical precipitation, and adsorption. Among these, adsorption is one of the most effective techniques for eliminating toxic substances using specialized adsorbents [120]. Adsorption, a widely used physicochemical method for removing different contaminants from water and wastewater, involves the accumulation of solute molecules at an interface, driven by mass transfer between the liquid phase and a solid-phase adsorbent. Adsorption can occur at various interfaces, including liquid–liquid, gas–liquid, gas–solid, and liquid–solid. The substance being adsorbed is called the adsorbate, while the material facilitating the process is known as the adsorbent [162]. Physical adsorption occurs due to intermolecular forces (van der Waals forces) between the adsorbent and adsorbate and can take place on any solid surface. Two types of physical adsorption are known: surface adsorption and electrostatic adsorption. Surface adsorption is influenced by the porosity and specific surface area of the adsorbent. A larger specific surface area enhances the physical adsorption effect. Smaller particle sizes lead to higher porosity, more active sites on the surface for pollutant adsorption, faster diffusion rates, and improved mass transfer of pollutants within the adsorbent [163]. In contrast, chemical adsorption involves the formation of chemical bonds between adsorbate molecules and the adsorbent through electron transfer, exchange, or sharing, often leading to chemical reactions. Key chemisorption mechanisms include reduction, surface coordination, chemical precipitation, and ion exchange. In water pollutant removal, steel slag undergoes both physical and chemical adsorption, with processes such as reduction, precipitation, coordination exchange, and ion exchange working synergistically to eliminate various contaminants [161].

Due to special qualities, high surface area and selectivity, magnetic zeolite NCs are one of the more effective adsorbents for water purification. Synthesized via co-precipitation, impregnation, or hydrothermal methods, these composites effectively remove persistent pollutants. Their key advantages include easy separation, reusability, and cost-effectiveness, making them particularly useful in regions lacking advanced water treatment technologies [23]. Figure 6 shows the removal of a particular pollutant from water by adsorption onto magnetic zeolite NCs.

Table 3 provides a summary of recent research on the absorption of different pollutants using magnetic zeolite NCs.

Meirelles et al. [123] developed a magnetic adsorbent by decorating the surface of FAU with magnesium ferrite (MgFe_2_O_4_) NPs, achieving a high specific surface area of 400 m^2^/g. The authors successfully removed about 94% of Co^2+^ and Mn^2+^ ions from aqueous solutions and retained more than 65% of its adsorption efficiency even after the third cycle of use. Phouthavong et al. [164] reported the synthesis of a BEA zeolite and Fe_3_O_4_ composite using the dry-gel conversion method and raw materials derived from agricultural waste. This magnetic composite demonstrated high efficiency in removing heavy metals from aqueous solutions. Chen et al. [165] showed that a magnetic composite material (Fe_3_O_4_-COOH@H-ZIF-67) with a hierarchical porous structure was synthesized and effectively used for removing benzimidazole pesticides from water via magnetic solid-phase extraction, showing high adsorption capacity, stability, and reusability. Magnetic Beta zeolite particles (~200 nm) were synthesized for potential use in targeted antitumor drug delivery, showing effective adsorption and controlled release of 5-fluorouracil in acidic conditions, biocompatibility with human blood cells, and biodegradability into non-toxic byproducts within 7 days [21]. Chen et al. [166] developed a novel, cost-effective method to synthesize high-performance magnetic zeolite 4A using kaolinite and red mud through a one-step activation–reduction and hydrothermal process. The resulting material, MZ-20 (with 20% red mud), exhibited a high adsorption capacity of 172 mg/g for Sr(II), removing 96.4% within 1 h and maintaining strong magnetic and regenerative properties over multiple cycles. Khatamian et al. [167] synthesized zeolite A with cubic morphology from Na-clinoptilolite and loaded with Fe_3_O_4_ and Fe_2_O_3_ NPs, where 3 wt.% magnetite and 1 wt.% maghemite showed optimal performance. The Fe_3_O_4_/zeolite A and Fe_2_O_3_/zeolite A NCs achieved up to 95.39% and 98.52% arsenic removal, respectively, with minimal NP agglomeration. Liu et al. [168] reports the synthesis of Fe_3_O_4_@ZIF-8, a magnetic core–shell adsorbent designed for efficient removal of organic pollutants like MB and DCF. The material showed rapid and high adsorption efficiencies (up to 98%) and outperformed conventional activated carbon, with thermodynamic and molecular modeling confirming strong, spontaneous adsorption mechanisms.

The efficient recovery and reuse of costly materials like catalysts play a crucial role in promoting sustainable process development [29]. Solid materials are widely recognized as effective heterogeneous catalysts due to their low-cost feedstocks and products, with the added expectation that these catalysts remain stable and fully recoverable. Among them, zeolites stand out for their remarkable thermal and chemical stability, structural versatility, and high shape selectivity, and they have been utilized across numerous applications. One major challenge is the separation of solid catalysts from viscous or semi-solid reaction mixtures, which often demand energy-intensive procedures, increasing both complexity and cost. To address this, incorporating magnetic properties into zeolites has emerged as an effective approach for facilitating their separation and reuse magnetic solid catalysts [169]. Figure 7 shows the use of magnetic zeolite NCs as a catalyst.

MRI is a widely used technique in medical diagnostics, yet its application is limited by inherently low sensitivity. The subtle differences in relaxation times between healthy and diseased tissues can make accurate diagnosis challenging. To enhance image contrast and improve diagnostic clarity, MRI contrast agents are employed. Among them, superparamagnetic iron oxide nanoparticles (SPIONs) serve as effective negative contrast agents. These NPs are typically coated with organic or inorganic materials to improve their stability and dispersion in biological environments. The type and properties of the coating significantly influence not only the biocompatibility and behavior of the contrast agent in the body but also its magnetic characteristics and relaxivity. Furthermore, achieving strong contrast enhancement in MRI with minimal administered doses is crucial to minimizing potential side effects. In this context, optimizing the amount of nano iron oxide incorporated into the zeolite structure is essential to ensure effective MRI contrast while maintaining biocompatibility [111]. Atashi et al. [20] reported on the synthesis of Fe_3_O_4_@ZSM-5 NCs, using a hydrothermal method to evaluate its potential as an MRI T_2_ contrast agent. The composite, with particle sizes ranging from 80 to 150 nm, showed high r_2_ relaxivity, good cytocompatibility, and promising imaging performance on a 1.5 T clinical MRI scanner. Charehaghaji et al. [111] also demonstrated that both the amount and dispersion of iron oxide NPs within the zeolite matrix play a crucial role in influencing MRI image contrast; the researchers synthesized Fe_3_O_4_/NaA NCs with varying iron oxide content to investigate their effect on MRI image contrast. They found that the NC with the lowest iron oxide content (3.4%) exhibited the highest T_2_ relaxivity and provided the best MRI image contrast.

Magnetic zeolite NCs are also very interesting for drug delivery because they can be activated by an external magnetic field [29]. Magnetically guided drug delivery systems, following the “drug-organ-target” principle, help achieve optimal local drug concentration while minimizing systemic toxicity through lower doses and prolonged retention at the target site. This is commonly achieved by incorporating MNPs into the drug carrier. These NPs typically clear from the bloodstream within 1–6 h post-injection, accumulating in organs before being fully eliminated [170]. Wu et al. [171] synthesized magnetic drug-loaded NPs using a Fe_3_O_4_@ZIF-8 NC, in which the zeolite imidazolate framework (ZIF-8) served as the outer shell, while Fe_3_O_4_ was encapsulated within. Doxorubicin (DOX) was loaded into the structure to enable drug delivery functionality. Targeted delivery of anticancer drugs investigated Brazovskaya et al. [21] too. Magnetic Beta zeolite was synthesized using a hydrothermal method. The composite effectively adsorbs and gradually releases the drug 5-fluorouracil, reaching a maximum release (45%) in an acidic environment (pH = 5.2), which is typical of inflammatory sites. At concentrations ranging from 0.1 to 10 mg/mL, it does not damage human blood cells, indicating its non-toxicity. NCs of magnetite and FAU zeolite with high specific surface area and adsorption capacity were prepared by mechanical activation at room temperature. In vitro tests showed that this material effectively stored and released DOX, making it a promising candidate for drug delivery applications [172]. Figure 8 schematically shows a magnetically guided drug delivery system using magnetic zeolite NCs.

Magnetic zeolite NCs have also been used to enhance the properties of different polymer matrices [29]. Murniati et al. [19] have shown in their work the impact of magnetically modified natural zeolite on the mechanical and damping properties of natural rubber reinforced with nanosilica. Using SIR 20 natural rubber and zeolite modified with Fe_3_O_4_ and a titanate coupling agent (TCA) as an alternative to traditional silane agents, they found significant improvements in mechanical performance. The enhanced properties were linked to increased crosslink density and the synergistic effect of the zeolite-nanosilica-magnetic filler blend, with or without a white oil softener. These findings suggest that magnetically modified zeolite-filled nanosilica can serve as a sustainable alternative to carbon black in damping applications.

Magnetic zeolite NCs have many other applications in addition to those presented above, depending on their properties and synthesis methods. The versatility of these materials lies in the ability to tailor their structure, surface functionality, and magnetic behavior during preparation [174].

## 7. Conclusions

Magnetic zeolite NCs represent a rapidly advancing class of hybrid materials that combine the structural and surface properties of zeolites with the magnetic functionality of NPs. This synergy enables multifunctionality while maintaining high porosity, ion exchange capacity, and structural stability. These features make magnetic zeolite NCs highly promising for a wide range of applications, including catalysis, environmental remediation, adsorption, drug delivery, and medical imaging.

This review has provided a comprehensive overview of the development of these materials—from the early stages of research to current synthesis techniques such as co-precipitation, hydrothermal synthesis, sol–gel, and others. Advanced characterization methods such as XRD, SEM, TEM, BET, FTIR, and VSM are critical in understanding the physicochemical properties and guiding the design of optimized composites.

The integration of MNPs into zeolite frameworks not only improves separation efficiency but also introduces novel functionalities without significantly compromising the structural integrity or catalytic performance of the zeolite. However, challenges remain in optimizing magnetic loading, ensuring long-term stability, improving biocompatibility, and scaling up production processes.

Future research should prioritize environmentally friendly synthesis methods, precise control over material architecture, and application-specific functionalization. Interdisciplinary collaboration will be essential to fully unlock the potential of magnetic zeolite NCs in both industrial and biomedical fields.

## Figures and Tables

**Figure 1 nanomaterials-15-00921-f001:**
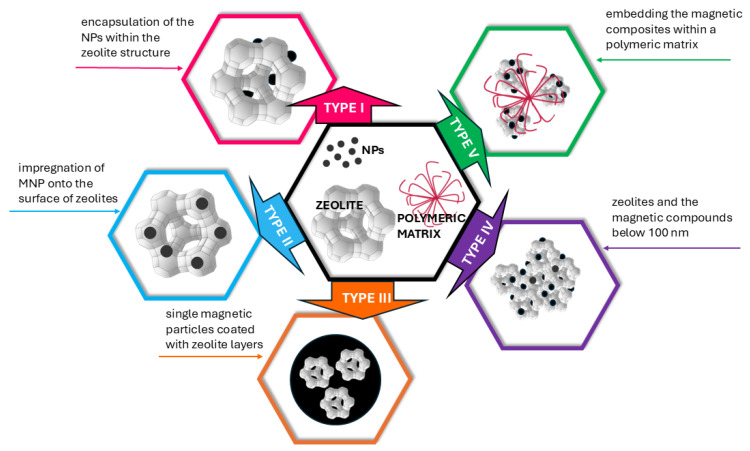
Five distinct types of magnetic zeolite NCs. Redrawn based on Ref. [29].

**Figure 2 nanomaterials-15-00921-f002:**
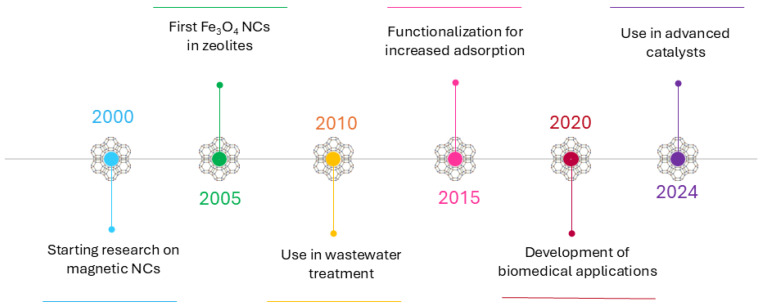
Evolution of research on magnetic zeolite NCs: from fundamental studies to advanced environmental and biomedical applications.

**Figure 3 nanomaterials-15-00921-f003:**
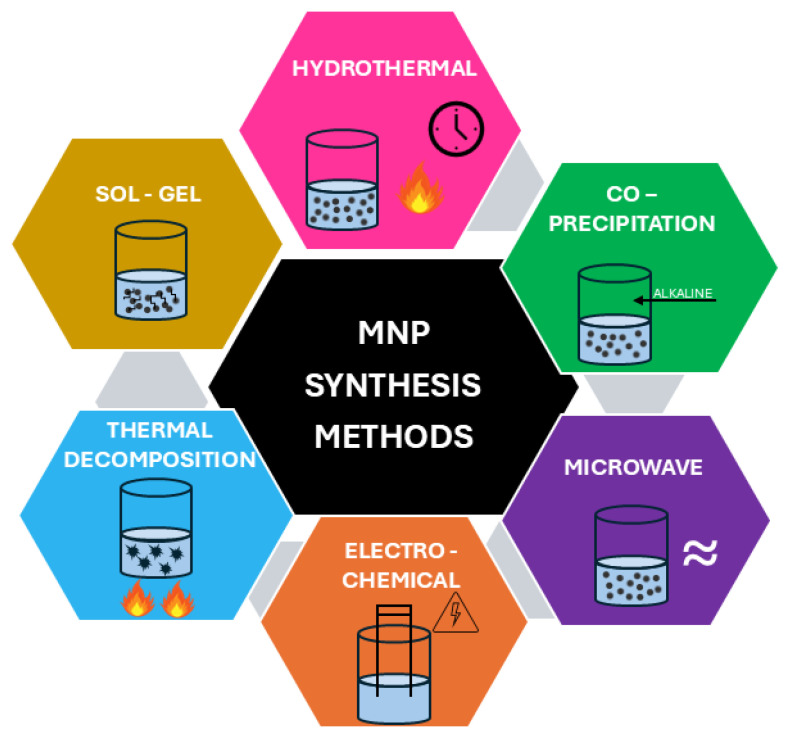
Different MNP synthesis methods.

**Figure 4 nanomaterials-15-00921-f004:**
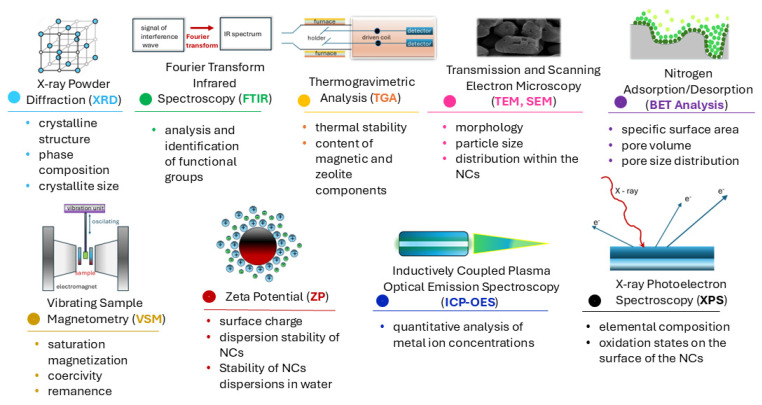
Schematic representation of the characterization methods for magnetic zeolite NCs.

**Figure 5 nanomaterials-15-00921-f005:**
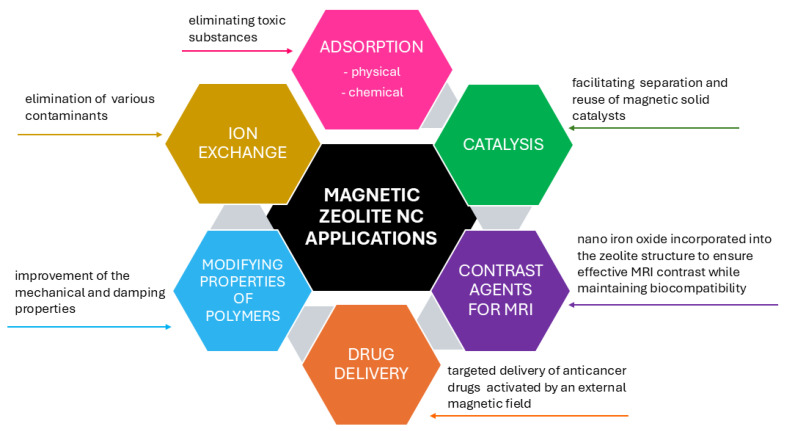
Magnetic zeolite NC applications.

**Figure 6 nanomaterials-15-00921-f006:**
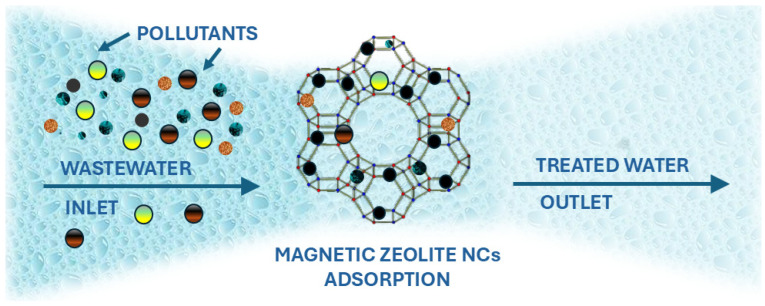
Adsorption-based water treatment. Redrawn based on Ref. [23].

**Figure 7 nanomaterials-15-00921-f007:**
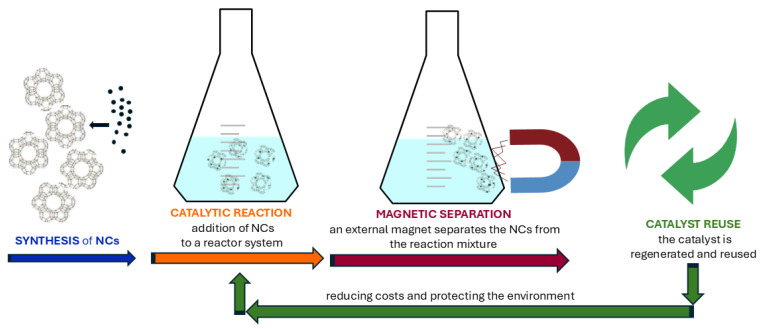
Magnetic zeolite NCs as catalyst.

**Figure 8 nanomaterials-15-00921-f008:**
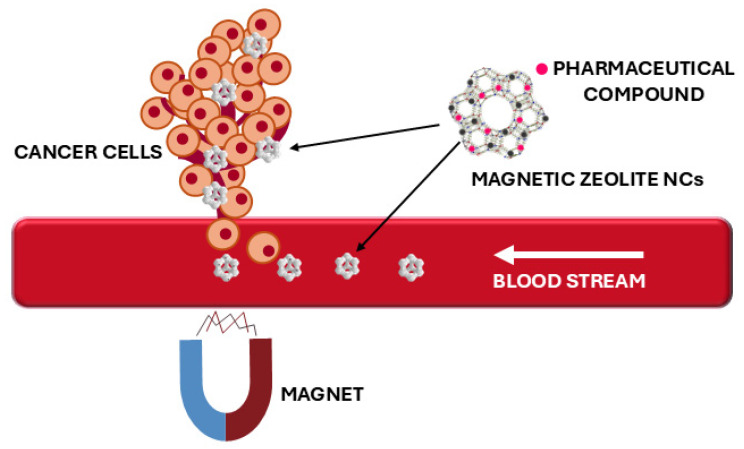
Magnetically guided drug delivery system using magnetic zeolite NCs. Redrawn based on Ref. [173].

**Table 1 nanomaterials-15-00921-t001:** Functionalization methods with main effects and application examples.

Type of Functionalization	Main Effect	Application
Ion Exchange	Selective adsorption of metals	Removal of heavy metals from wastewater [4]
Acid Treatment	Increased specific surface area	Catalysis, dye adsorption [38,39]
Alkaline Treatment	Formation of mesopores	Removal of large organic molecules [40,41]
Silanization	Hydrophobicity/hydrophilicity	Separation of oil pollutants [42]
Metal Oxides	Magnetic/photocatalytic properties	Water purification, catalysis [43]
Polymer Coating	Stabilization, dispersion	Biomedical applications [44]

**Table 2 nanomaterials-15-00921-t002:** The most commonly used synthesis methods for MNPs with their advantages and limitations.

Synthesis Method	Description	Advantages	Limitations	References
Co-precipitation	MNPs are prepared from aqueous salt solutions, by the addition of a base at room temperature or at high temperatures. By selecting the type of salt, the Fe^3+^/M^2+^ stoichiometric ratio, temperature, and pH value, we can significantly influence the size, shape, and composition of the particles.	• Simple and fast • Low reaction *T* • Environmentally friendly solvent • High and scalable reaction product	• Narrow size distribution • Poor size control	[39,124,125]
Microemulsion	Within the water droplets of one reverse microemulsion, there is a solution of metal ions, while in the water droplets of another reverse microemulsion, there is a solution of the precipitating reagent. Upon collision, the micelles merge, bringing the reactants into contact and allowing them to react and form a product. This is followed by nucleation and the growth of the newly formed particles.	• Use of simple equipment • Product with crystalline structure and high specific surface area • Simple conditions of synthesis • Efficient control over size, shape, and composition • High monodispersity	• Low yield • Limited thermal stability • Complex in multi-component systems	[124,126,127,128]
Thermal decomposition	High-temperature decomposition of organometallic precursors in high-boiling organic solvents containing stabilizing surfactants.	• Produces nearly monodispersed NPs • Precise control over size and shape • Narrow size distribution	• High temperature • Long reaction time • Low yields	[39,124,129,130]
Hydrothermal	Includes various wet-chemical techniques for crystallizing materials in a sealed container from an aqueous solution at high temperatures (130 °C to 250 °C) and elevated vapor pressures (0.3 to 4 MPa).	• Grows crystals of many different materials • Various morphologies can be achieved by varying the synthesis temperature • Ability to yield uniform particles with excellent magnetic properties	• Size, shape, and crystal structure of NPs are highly dependent on synthesis conditions • High pressure	[39,124,131]
Sonochemical	Ultrasound induces cavitation to create extreme reaction conditions, including high temperatures, pressures, and cooling rates.	• Controlled size and shape • pH and ultrasound intensity can influence particle morphology and crystallinity	• Careful control of reaction parameters	[124,132]
Microwave-assisted	The reagents absorb microwave energy, leading to uniform heating and a rapid chemical reaction.	• Fast reaction time • Controlled size and properties • Monodispersed particles • High yields • Energy-efficient method • Minimal processing steps	• Careful control of reaction parameters	[133,134]
Sol–gel	This method involves hydroxylation and condensation of molecular precursors, forming a “sol” of NPs. Further polymerization creates a 3D metal oxide network (wet gel), requiring heat treatment for crystallization.	• Good control of the particle size • Controllable microstructure and homogeneity	• Pollution from byproducts of reactions • Need for post-treatment of the products	[124,135]
Electro-chemical	By using electric current, metal ions in solution are reduced, leading to the formation of NPs on the electrode.	• Environmentally friendly • Cost-effective method • Controllable process	• Use of organic solvents	[136,137]

**Table 3 nanomaterials-15-00921-t003:** Summary of recent studies on the adsorptive removal of pollutants using magnetic zeolite NCs.

Pollutant	Matrix	MNPs	Zeolite	Removal Efficiency	Ref.
Co^2+^, Mn^2+^	Aqueous solutions	MgFe_2_O_4_	Faujasite (FAU)	94%	[123]
Heavy metals (Pb^2+^, Cu^2+^, Zn^2+^)	Aqueous solutions	Fe_3_O_4_	BEA	70–90%	[164]
Benzimidazole pesticides (BZD)	Simulated pesticides wastewater	Fe_3_O_4_-COOH	H-ZIF-67	82.76–96.18%	[165]
5-Fluorouracil antitumor drug	Human blood cells	Fe_3_O_4_	BetaBeta	45%	[21]
Sr^2+^	Radioactive wastewater	Fe_3_O_4_	4A	96.4%	[166]
As	Contaminated water	Fe_2_O_3_/Fe_3_O_4_	A	95.39%/98.52%	[167]
Methylene blue (MB) and Diclofenac sodium (DCF)	MB and DCF water solution	Fe_3_O_4_	Zeolite Imidazolate Framework-8 (ZIF-8)	98%	[168]

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
