# Peer review of "Synthesis, Characterization, and Application of Magnetic Zeolite Nanocomposites: A Review of Current Research and Future Applications"

_nanomaterials, 2025, doi:10.3390/nano15120921_

Round 1

Reviewer 1 Report

Comments and Suggestions for Authors

This manuscript presents a literature review on the synthesis, characterization and applications of zeolite nanocomposites with magnetic nanoparticles. The review is well structured, but I believe it needs some adjustments to improve the work presented.
1. In characterization techniques, the limitations of the techniques presented should also be indicated.
2. In the particular case of nitrogen adsorption/desorption, it is necessary to note that BET analysis is not recommended for microporous materials such as zeolites, and other techniques for analyzing the results should be chosen. This particular part needs to be reviewed.
3. The part relating to applications focuses on water treatment, presenting gaps in other applications. This aspect should be rethought to strengthen other applications. Alternatively, focus the review on the application in water treatment.

Author Response

Dear Reviewer,

we sincerely thank you for your time and effort in reviewing our manuscript entitled »Synthesis, Characterization, and Application of Magnetic Zeolite Nanocomposites: A Review of Current Research and Future Applications«.

We are grateful for your insightful comments and constructive suggestions, that have significantly contributed to the improvement of our work.

This manuscript presents a literature review on the synthesis, characterization and applications of zeolite nanocomposites with magnetic nanoparticles. The review is well structured, but I believe it needs some adjustments to improve the work presented.

COMMENT 1:

In characterization techniques, the limitations of the techniques presented should also be indicated.

RESPONSE 1:

We sincerely thank the Reviewer for this valuable suggestion. We fully agree that a critical discussion of the limitations of individual characterization techniques is important for providing a balanced and comprehensive overview. As advised, we have carefully revised the corresponding section and added specific remarks on the limitations of each technique, including XRD, FTIR, TGA, SEM/TEM, BET, VSM, ZP, ICP-OES, and XPS. These additions aim to highlight potential constraints such as surface sensitivity, interpretative challenges, sample preparation issues, and suitability for certain material types (e.g., microporous structures in BET analysis). We believe that these updates strengthen the scientific rigor of the manuscript and improve its clarity for the readers. The revised text is highlighted in yellow in the manuscript for easy identification by the reviewer.

COMMENT 2:

In the particular case of nitrogen adsorption/desorption, it is necessary to note that BET analysis is not recommended for microporous materials such as zeolites, and other techniques for analyzing the results should be chosen. This particular part needs to be reviewed.

RESPONSE 2:

We thank the reviewer for this valuable comment. We agree that BET analysis has limitations when applied to microporous materials such as zeolites. In response, we have revised the section discussing nitrogen adsorption/desorption to clarify that while BET can be used to estimate surface area, it may lead to inaccuracies in microporous systems. We have also mentioned alternative approaches such as the t-plot method or density functional theory (DFT) models, which are more appropriate for evaluating microporosity. The revised section now more accurately reflects the limitations and recommended practices for analyzing the textural properties of zeolite-based nanocomposites.

COMMENT 3:

The part relating to applications focuses on water treatment, presenting gaps in other applications. This aspect should be rethought to strengthen other applications. Alternatively, focus the review on the application in water treatment.

RESPONSE 3:

We thank the reviewer to this comment. We agree that there is a slightly greater emphasis and review of recent publications in the field of water purification and treatment using magnetic zeolite NCs, but with good reason, as water purification is one of the most promising and widespread applications. This is due to their cost-effective and environmentally friendly nature compared to other approaches, which is also reflected in the growing number of scientific publications on the topic. This explanation is also included and highlighted in yellow. However, all other applications of magnetic zeolite NCs are also described, clearly illustrated with visual material, and for each of them, the most recent or significant findings are highlighted.

Reviewer 2 Report

Comments and Suggestions for Authors

This review highlights the synthesis methods, key characterization techniques, and diverse applications of magnetic zeolite NC in fields such as adsorption, catalysis, drug delivery, and MRI. By detailing their structural features, interaction modes, and advancements in research, this review aims to inspire innovative applications and provide a foundation for future studies on magnetic zeolite NCs. Overall, this is comprehensive review for applications of magnetic zeolite nanocomposites, emphasizing their potential in environmental and biomedical fields. I recommend this review to publish in Nanomaterials after revision. Herein, I provide some suggestions for the authors to improve the manuscript.

  1. It is important to clearly state the purpose of this review at the end of the introduction to guide readers on its focus and objectives. Including a concise statement will help emphasize the significance of the topic and the intended contributions of the review to the field.
  2. The caption of Figure 1 should clearly describe the five distinct types of magnetic zeolite NCs, emphasizing their unique structural features and functional properties. Providing such details in the caption will enhance the reader's understanding and make it easier to differentiate between the various types of NCs discussed.
  3. The caption for Figure 5 should comprehensively outline the specific applications of magnetic zeolite NCs. Providing detailed information will help readers understand the practical significance of these NCs and their relevance to various fields.
  4. In the introduction “Nanotechnology has garnered global attention, with significant efforts directed to…”, the sentence effectively highlights the global importance of nanotechnology, but additional references to recent advancements or key studies would strengthen the statement and provide more credibility.

https://doi.org/10.1186/s12951-023-02208-3

Author Response

Dear Reviewer,

we sincerely thank you for your time and effort in reviewing our manuscript entitled »Synthesis, Characterization, and Application of Magnetic Zeolite Nanocomposites: A Review of Current Research and Future Applications«.

We are grateful for your insightful comments and constructive suggestions that have significantly contributed to the improvement of our work. We have carefully considered each of your comments and made the necessary revisions accordingly. Below, we address your valuable feedback point by point.

This review highlights the synthesis methods, key characterization techniques, and diverse applications of magnetic zeolite NC in fields such as adsorption, catalysis, drug delivery, and MRI. By detailing their structural features, interaction modes, and advancements in research, this review aims to inspire innovative applications and provide a foundation for future studies on magnetic zeolite NCs. Overall, this is comprehensive review for applications of magnetic zeolite nanocomposites, emphasizing their potential in environmental and biomedical fields. I recommend this review to publish in Nanomaterials after revision. Herein, I provide some suggestions for the authors to improve the manuscript.

COMMENT 1:

It is important to clearly state the purpose of this review at the end of the introduction to guide readers on its focus and objectives. Including a concise statement will help emphasize the significance of the topic and the intended contributions of the review to the field.

RESPONSE 1:

We sincerely thank the Reviewer for this helpful comment. In response, we have revised the end of the Introduction to include two sentences that clearly outline the purpose, focus, and structure of the review. This addition is intended to guide readers and emphasize the significance and contributions of the article to the field.

COMMENT 2:

The caption of Figure 1 should clearly describe the five distinct types of magnetic zeolite NCs, emphasizing their unique structural features and functional properties. Providing such details in the caption will enhance the reader's understanding and make it easier to differentiate between the various types of NCs discussed.

RESPONSE 2:

Thank you for your valuable suggestion. In response, we have revised Figure 1 by adding a detailed description of the different types of magnetic zeolite NCs directly within the image to enhance clarity and aid the reader’s understanding of their structural and functional distinctions.

COMMENT 3:

The caption for Figure 5 should comprehensively outline the specific applications of magnetic zeolite NCs. Providing detailed information will help readers understand the practical significance of these NCs and their relevance to various fields.

RESPONSE 3:

Thank you for your thoughtful suggestion. The methods presented in Figure 5 are indeed extensive and are thoroughly explained in the accompanying text. Therefore, it would be challenging to include all details directly in the figure caption. Nevertheless, we have incorporated several key application outlines into the figure itself to improve clarity and provide readers with a concise visual summary.

COMMENT 4:

In the introduction “Nanotechnology has garnered global attention, with significant efforts directed to…”, the sentence effectively highlights the global importance of nanotechnology, but additional references to recent advancements or key studies would strengthen the statement and provide more credibility.

RESPONSE 4:

Thank you for this helpful and constructive suggestion. To strengthen the statement and provide a more credible foundation, we have added a recent and relevant reference that highlights current advancements in the field of nanotechnology. This citation supports the remark regarding the global impact and increasing research efforts in this area. The updated reference has been incorporated into the Introduction section of the manuscript.

Round 2

Reviewer 1 Report

Comments and Suggestions for Authors

The authors reply to all my concerns.